# Exploring diabetes status and social determinants of health influencing diabetes-related complications in a Northwestern community, Ontario, Canada: A mixed method study protocol

**Idevania G. Costa**[1,2,3]*, **Kristen McConell**[1,2], **Kaitlin Adduono**[1,2], **Pilar Camargo-Plazas**[4], **Anna Koné**[1,3,5,6,7]

**1** Department of Health Sciences, Lakehead University, Thunder Bay, ON, Canada, **2** School of Nursing, Lakehead University, Thunder Bay, ON, Canada, **3** Centre for Education and Research on Aging and Health (CERAH), Thunder Bay, ON, Canada, **4** School of Nursing, Queen's University, Kingston, ON, Canada, **5** Behavioural Research and Northern Community Health Evaluative Services (Branches), Thunder Bay, ON, Canada, **6** Health System Performance Network (HSPN), Toronto, ON, Canada, **7** Centre for Rural and Northern Health Research, Sudbury, ON, Canada

* igcosta@lakeheadu.ca

## Abstract

Diabetes is a common chronic condition affecting the many spheres of individuals' lives. It can also lead to severe complications without continuous management. Accordingly, this paper describes a study protocol aimed at 1) determining the status and prevalence of diabetes complications in a Northwestern Ontario community; 2) exploring the internal (e.g., demographic and clinical variables) and external factors (e.g., access to services and resources) affecting diabetes outcomes (e.g., complications, emergency room visits, hospitalizations); 3) critically exploring how the social determinants of health affect self-management for individuals living with diabetes; and 4) identifying individuals' needs, concerns, and challenges to monitor and regulate diabetes. The study uses a cross-sectional design and a social constructivist approach based on qualitative data collection. The proposed study will include patients with type 1 and type 2 diabetes with or without diabetes complications who have been attending the Centre for Complex Diabetes Care (CCDC) in Thunder Bay, Ontario, Canada, since January 2019. Quantitative data related to diabetes complications and other outcomes, diabetes management, and demographic and clinical status will be retrieved from patients' charts using a data extraction form. Analyses of the quantitative data will include the prevalence of diabetes complications, rate of hospitalizations, and their associations with diabetes management, access to services, and social determinants of health. Additionally, interviews will occur with at least 10 participants with or without diabetes complications to understand their needs, concerns, and struggle to self-manage diabetes daily. The results of this study will generate evidence to support future research and policy on the development and implementation of an educational program to improve self-care management and outcomes for individuals living with diabetes and its complications in Northwestern Ontario.

**Data Availability Statement:** No datasets were generated or analysed during the current study. All relevant data from this study will be made available upon study completion.

**Funding:** This research was funded by Lakehead University's Senate Research Committee- CIHR Research Development Fund The funders had and will not have a role in study design, data collection and analysis, decision to publish, or preparation of the manuscript.

**Competing interests:** The authors have declared that no competing interests exist.

## Introduction

Diabetes has become one of the most costly and challenging chronic diseases of our time, affecting approximately 422 million people worldwide [1]. In Canada, 11 million people have diabetes or prediabetes [2]. Over 1.6 million people live with diabetes in Ontario, and another 2.3 million more live with pre-diabetes [3]. Type 2 diabetes is the most prevalent type of diabetes and accounts for 90 to 95 per cent of all diabetes cases [4]. To make matters worse, people living with diabetes are susceptible to several types of complications considered life-threatening. A significant number of complications, including strokes (30 percent), heart attacks (40 percent), kidney failure requiring dialysis (50 percent), and non-traumatic amputations (70 percent) affect Canadians with diabetes each year [3, 5].

People living with diabetes in the Northwest Ontario (NWO), particularly those living in rural and remote communities, are facing greater negative outcomes of diabetes such as emergency visits, and hospitalization rates as a result of food insecurity, lack of support, and scarcity of healthcare services and resources, which increase the risk of complications of diabetes [6, 7]. Sadly, the North West Local Health Integration Network (LHIN) has reported significantly higher rates (1135 per 100,000 diabetes population) of hospitalizations related to diabetes complications in NWO compared to the rest of the province (313 per 100,000 diabetes population) [8]. Worse, the rate of lower-limb amputation for people living in NWO communities, particularly the rural First Nation communities, has been reported to be seven times the provincial rate [9]. Furthermore, the rate of advanced kidney disease related to diabetes in NWO communities was double the rate of that in the general population between March 2014 and May 2017 [10]. In a recent study, authors reported the prevalence of diabetes for the total NWO population as 15.1 percent versus a Canadian prevalence of 8.8 percent [5]. Overall, diabetes comes at high cost. In 2019, the cost for treating diabetes in Canada was $30 billion [11], and the cost of diabetes-related complications such as diabetic foot ulcers has been estimated as $23,000 per patient per year [12].

Given that the poor outcomes of diabetes significantly impact the physical, psychological, and social areas of individuals' lives, more research is needed to understand and address the factors leading to poor diabetes outcomes. To our knowledge, limited published studies exist related to diabetes in NWO communities [4, 5, 8]. Also, the influence of social determinants of health on diabetes or diabetes-related complications, particularly the impact of gender, race/ethnicity for individuals living with diabetes in NWO have not been fully studied [13–15].

Evidence exists that Canadians with diabetes living in remote and rural areas of NWO face many challenges in addressing the social and economic determinants of health, particularly food insecurity and access to services and resources to support their diabetes management [16–18]. Food insecurity is associated with an increased risk of compromised management of type 1 and type 2 diabetes through several mechanisms [16, 18].

Social determinants of health represent an important factor, especially for population groups living in rural and remote areas. These populations usually experience food insecurity and difficulty in accessing healthcare services due to financial constraints and low income [15, 17, 18]. Therefore, it is crucial to consider this specific context when evaluating diabetes outcomes and the needs and concerns for people with diabetes in remote communities, which contribute to improve patients' outcomes and quality of life [14]. Such focus is particularly relevant and urgently needed for people with type 2 diabetes in rural and remote areas of NWO.

Researchers also suggest that health literacy, diabetes knowledge, and social support—as social determinants of health—are associated with improved clinical outcomes, reduced economic and psychosocial burden, and improved adaptation to better lifestyle activities [14, 15, 19]. While a few studies address social determinants of health for people living with diabetes in

rural and remote NWO [7, 9, 10, 20], the relationships between socioeconomic factors and individual attributes such as gender and race/ethnicity have not been explicitly studied in the context of a population living in NWO, Canada. Thus, it is necessary to explore the relationship between attributes such as gender, race/ethnicity, and social determinants of health influencing diabetes-related complications in order to address the increased burden of diabetes in NWO communities.

Such exploration is necessary because the needs, concerns, values, and struggles faced by NWO communities living with diabetes and diabetes complications are yet to be described and published. Such knowledge may help support the institutions providing diabetes care in NWO to develop strategies to improve diabetes and diabetes-related outcomes, enhance self-management skills, wellness, health accessibility, and equitable care among people living with diabetes in NWO.

To help close this gap this study aims to: 1) identify the status and prevalence of diabetes complications; 2) investigate the association between internal (e.g., socio-economic-cultural and clinical data, self-management skills and knowledge, health literacy, support, self-efficacy, motivations) and external factors (e.g., access to services and resources) on diabetes outcomes (e.g., complications, hospitalization, diabetes management); 3) critically explore how the social determinants of health affect self-management for individuals living with type 1 and type 2 diabetes; and 4) explore individual's needs, concerns, and challenges in regulating and self-managing diabetes.

## Methods

This research will use a parallel mixed-method approach to help us to gain a comprehensive picture of the diabetes status, outcomes, and individuals' needs and concerns while they self-manage their condition. A mixed-method approach was chosen in order to combine the benefits of both quantitative and qualitative methods by using numbers and narratives that will better reflect individuals' experiences; and hence, help us achieve the four research objectives [21]. In addition, collecting diverse types of data creates data triangulation and helps us to have greater insights on the diabetes status, care, needs, and challenges faced by individuals in NWO. It also provides more valid and stronger inferences than a sole method would achieve [22].

Thus, to achieve the study's objectives, this study will first use a quantitative cross-sectional design to assess prevalence and determinants of diabetes-related complications (objectives 1 and 2). Secondly, it will employ a qualitative research design that uses a social constructivist approach informed by Charmaz [23] to explore the needs, concerns, values, and struggles faced by people in NWO communities living with diabetes and diabetes complications and how the social determinants of health affect diabetes self-management for this population (objectives 3 and 4).

### Research setting

Qualitative and quantitative data for this study will be retrieved from the Centre for Complex Diabetes Care (CCDC) located in Thunder Bay. The CCDC currently has approximately 900 active patients with two locations to serve NWO communities in Thunder Bay and rural areas, one in Thunder Bay and the other in Sioux Lookout. It supports patients living with diabetes-related complications and collective health issues that require more specific treatment strategies to avoid further complications. The CCDC program is available to both inpatients and outpatients on a referral basis. People with diabetes from the Thunder Bay region access the CCDC through a referral from their primary healthcare provider or specialist. The CCDC

team works directly with patients to ensure comprehensive care; and provides level-three care to 15 percent of the diabetes population. In level-three care, people require intensive care management for their chronic disease and associated complications [24]. Telemedicine is used whenever possible, making patients stay close to home with the option for in-person appointments with a Registered Nurse, Registered Dietician, Nurse Practitioner, Physician, and Social Worker [24]. Considering that the clinic cares for people with different diabetes experiences, management needs, and education, we believe this patient population is well-suited for the study goal.

## Study design

**Part 1: A quantitative prevalence study.** *Data source and variables*. The data for this retrospective study will be collected from patients' charts through the health records department of the Thunder Bay Regional Health Sciences (TBRHSC). The data is routinely collected as part of the care management of patients enrolled at CCDC program. For this study the charts of participants assessed and followed up by the CCDC since January 2019 will be reviewed. The research team will utilize a data-collection form (Table 1) to request socio-demographic and healthcare information, including age, gender, employment, marital status, education, living arrangement, household income, and access to services and resources.

Clinical and care management data such as duration of diabetes, types of diabetes, hemoglobin a1c, current diabetes treatment regimen, self-management strategies, and confidence to manage diabetes will be included. Outcome data related to diabetes complications, types and number of complications, and hospitalizations due to diabetes or its complications will also be retrieved using the data collection form. The data collection form will be piloted with five participants to ensure it has alignment with the data available in their charts. Then the team will meet to make necessary amendments to the data collection form. All data will be encrypted using an anonymized identification number and will be stored and analyzed in a secure environment at Lakehead University. Only the principal investigator and authorized research assistants will have password access to the computer and secured folder on the H drive.

*Study population*. Participants will be included if they meet the following criteria: 1) age 18 or more, 2) diagnosed with any type of diabetes, 3) with or without diabetes complications, and 3) enrolled with the CCDC for at least six months and assessed or followed up by the clinical staff between January 2019 and December 2022. Those meeting these criteria but who are not active (i.e., not attending appointments) during the study period will be excluded from the study. Complications and hospitalizations among these patients will also be evaluated during the same period, as indicated on the patient's medical chart. Approximately 300 patients' charts will be assessed to retrieve the data in Table 1. This number is based on the information from the CCDC manager that state approximately 300 active patients in the program.

*Data analysis*. All the collected data will be organized in a secure Excel spreadsheet and then transferred to SPSS software (version 25) for statistical analysis. Analyses will include descriptive statistics to evaluate demographic and clinical characteristics of the study population as well as prevalence of diabetes-related complications. We will use bivariate analyses (t-tests, crude OR) and multivariate logistic regression (adjusted odd ratios) to test the associations between the risk factors and diabetes complications and hospitalizations. A p-value $< 0.05$ will be considered to be statistically significant. All the results will be presented in an aggregated format and small cells ($<6$) will be deleted to avoid possible re-identification.

**Part 2: Social constructivist approach.** *Qualitative data collection*. An interview guide with open-ended questions related to needs, experiences, values, struggles, and contextual factors affecting individuals' everyday diabetes self-care management practices has been

**Table 1. Demographic and clinical information (for interview and patient's chart).**

| Demographic information | | |
|---|---|---|
| **Variables** | **Response** | **Note** |
| **Demographic Variables** | | |
| Age (years) | | |
| Gender | F () M () Other: | |
| Education (N. of years) | None (if applicable) | |
| | 1 to 4 | |
| | 5 to 8 | |
| | 9 to 12 | |
| | Post secondary (College or university) incomplete | |
| | Post secondary (College or university) Complete | |
| Marital status | Single | |
| | Married/partnered | |
| | Divorced | |
| | Widowed status | |
| | Other: | |
| Social support: | Spouse | |
| | Relative | |
| | Friend | |
| | Other: | |
| Living arrangement | Living with family | |
| | Living alone | |
| | Other: | |
| Employment status | No | |
| | Yes | |
| | Working at home | |
| | Retired | |
| | ODSP Benefit or other | |
| Household income (CND$) | Under $30,000 | |
| | $30,000 to $59,999 | |
| | $60,000 to $ 89,999 | |
| | $\geq$ 90,000 | |
| **Clinical Variables** | | |
| Diabetes type | Type 1 | |
| | Type 2 | |
| | Other | |
| | Type 3 | |
| Diabetes duration (years) | | |
| HbA1C (value) | | |
| BMI | $\leq$25 | |
| | >25 | |
| Active smoke | Yes | |
| | No | |
| Number of cigarettes/day | $\leq$1 | |
| | $\geq$1 | |
| Smoke history | Yes | |
| | No | |

*(Continued)*

**Table 1.** (Continued)

| Variables | Response | Note |
|---|---|---|
| **Demographic information** | | |
| Diabetes treatment regimen | Diet only | |
| | Oral agent only | |
| | Insulin only | |
| | Insulin and oral agent | |
| | Injectable & oral agent | |
| Comorbidities | Heart disease | |
| | Respiratory disease | |
| | Kidney disease | |
| | Eye disease (diabetes related) | |
| | Others: | |
| **Access to services and resources** | | |
| Service | Family practitioner (Physician, NP) | |
| | Diabetes educator | |
| | Dietitian | |
| | Endocrinologist | |
| | Other | |
| Resources | Oral medication for diabetes | |
| | Insulin | |
| | Glucose meter and sensor | |
| | Other | |
| Diabetes outcome | | |
| Diabetes complication type | Kidney disease | |
| | Retinopathy | |
| | Neuropathy | |
| | Cardiovascular disease | |
| | Amputation | |
| | Other | |
| Duration of diabetes complication | 2–4 | |
| | 5–6 | |
| | 7–9 | |
| | 10–12 | |
| | 13–15 | |
| | $\geq 16$ | |
| Number of complications | 1 | |
| | 2 | |
| | 3 | |
| | 4+ | |
| Emergency room visits due to diabetes issues | 1x | |
| | 2x | |
| | 3x | |
| | $\geq 4$ | |
| Hospitalization due to diabetes complications | 1x | |
| | 2x | |
| | 3x | |
| | $\geq 4$ | |

developed. This interview guide will be used for the qualitative approach component of this study (Table 2). The data sources of this component of the study include interviews in person (if participants feel comfortable with this option) and with all measures to prevent COVID-19 infection. Participants will also have the option of choosing interview either over the phone or via Zoom. Field notes and researcher' journal will be used as a method of data triangulation. The semi-structured interview guide outlined in Table 2 provides direction, focus, and clarity for exploring people's experiences, needs, concerns, and challenges in monitoring and regulating diabetes.

The semi-structured interview guide will be pilot tested by the first (IGC) and third authors (KA) with three participants attending the CCDC and accepting participation in the study. Then the team will meet to make necessary amendments to the interview guide and the third author will continue to conduct the interviews.

*Study population*. Eligible participants for the qualitative component of this study will be recruited with the support of the CCDC team. Potential participants are those using the CCDC program, diagnosed with type 1 or type 2 diabetes, and who may or may not be experiencing a complication in the last six months, with the support of the knowledge user (stated above) through a purposive sample. The complications may include microvascular (kidney disease, retinopathy, and neuropathy) and macrovascular (cardiovascular) disease. Participants in this study will be eligible to participate if they meet the following criteria: 1) are 18 years or older; 2) have a confirmed medical diagnosis of diabetes for at least three years and with level three support, which ensures that participants can reflect on their experience of living and managing diabetes daily; 3) are able to speak and read in English comfortably and articulate their experience of having and managing diabetes; 4) are willing to engage in active self-reflection about their experience living with and managing diabetes; and 5) agree to participate by providing telephone consent. We will exclude women with gestational diabetes, people with pre-diabetes, a new diagnosis of diabetes, and a recent diagnosis of diabetes after an organ transplant, and people with mental illness with a diagnosis of diabetes who are not able to articulate their experience and journey of managing the disease.

The number of participants recruited in this study will be guided by the principle of saturation, which occurs when themes and categories in the data become repetitive and redundant such that no new information can be gathered by further data collection [25]. We anticipate between 10 to 20 participants.

*Qualitative analysis*. Interviews will be transcribed and stored within NVivo qualitative analysis software that will be password protected. The data analysis will follow an interpretive thematic analysis based on social constructivist and symbolic interactionism approaches [23, 26]. The interviews will be read first as a whole, second as sections of descriptions and stories, and third as phrases and words to identify specifically informative statements or themes [27]. The data will be organized and analyzed around crucial emerging themes of individuals with and without diabetes to help identify motivations, actions, challenges, and unique needs related to participants' experience caring for their diabetes. Our strategies to address trustworthiness of the study are informed by Tracy [28] and presented in Table 3.

After completion of a separate analysis of each quantitative and qualitative study, we will combine findings from each wing of the study to analyze how complementary or contradictory they are. Additionally, we will examine how interview findings could improve our interpretation of the statistical analysis.

## Ethical aspects

This study protocol has received approval by the Research Ethics Board at TBRHSC (TBRHSC # 2020526) for both quantitative and qualitative parts. Considering that the quantitative

**Table 2. Semi-structured interview guide.**

| Diabetes questions: |
|---|
| 1. How did you find out that you have diabetes? |
| Follow-up question: What were your symptoms when you were first diagnosed? |
| 2. What does it mean to you to have diabetes? |
| Probe: What have you learned from having it? |
| 3. How do you take care of your diabetes daily? |
| 4. When I say "needs for diabetes care" what does this mean to you? |
| 5. When are your needs for diabetes care? |
| Follow-up question: How have your needs been met? |
| 6. When I say "preferences for diabetes care," what does this mean to you? If participants ask, definition of preferences for diabetes care: Preferences represent your choices or priorities derived from your values for care. There are no right or wrong answers to preferences for your diabetes care. |
| 7. What are your preferences for diabetes care? |
| Follow-up question: How do they coincide (or overlap) with other parts |
| of your life? |
| 8. When I say "values for diabetes care," what does this mean to you? |
| If participants ask, definition of values for diabetes care: Values represent your ideal or best diabetes care. In other words, what is most important to you in your diabetes care. There are no right or wrong answers to your values for diabetes care. |
| 9. What are your values for diabetes care? |
| Follow-up question: How do they coincide (or overlap) with other parts of your life? |
| 10. What services do you access for diabetes care and what is the distance (if any) that you have to travel to access them? |
| 11. Do you feel that your doctors or other members of your diabetes health care team (e.g., diabetes educator, dietitian, and nurse practitioner) address your needs, preferences and values for care? If yes, please explain. |
| 11. What would you change about your diabetes care if you could? |
| 12. Is there anything else you would like to share about your values and preferences for diabetes care? |
| 13. When I say "challenges to regulate diabetes," what does this mean to you? |
| 14. What are the challenges you have faced to regulate your diabetes? |
| Follow-up question: How do they coincide (or overlap) with other parts |
| of your life? |
| 15. Tell me about access to services to manage your diabetes? |
| Follow-up question: What are the specialists you have seen so far?(e.g., diabetes educator, dietician, endocrinologists) |
| 15. Tell me about resources (diabetes supplies) to manage your diabetes? |
| Follow-up question: Describe your access to supplies. |
| 16. Tell me about diabetes education to manage your diabetes? |
| Follow-up question: How often you have attended diabetes classes? How did it help you to manage diabetes? |
| Diabetes complications questions: |
| 1. Tell me about your diabetes complication (e.g., kidney, eyes, heart or feet problem). |
| 2. Follow up question: Tell me about the first time you found you had a diabetes complication. How did you react? |
| 3. How did you react? Did someone talk to you about diabetes complication when you were first diagnosed with diabetes? |
| 4. What did you know about diabetes complications before having it? |
| 5. What do you think contributed to develop this complication you have? |
| 6. What does it mean for you to have this diabetes complication? |
| Probe: How do you view it? |
| 7. How do you take care of it on a daily basis? |
| 8. Tell me about access to services to manage your diabetes complication before you enrolled with CCDC? |

(*Continued*)

**Table 2.** (Continued)

| |
|---|
| Probe: what are the challenges and struggle you have found? |
| 9. Tell me about referral to diabetes specialists |
| 10. Tell me about access to resources (e.g., supplies) to manage your diabetes complication? |
| Probe: what are the challenges and struggle you have found? |
| 11. Tell me about your social support (e.g., family member, friends etc)? |
| Probe: Tell me more about that. |
| 12. How do you see your participation in your own care? |
| Probe: How comfortable do you feel. Please give example? |
| 13. Is there anything you would like to add that I might have missed? |
| 14. Do you have any questions? |

portion of this study involves retrospective chart review only, in which no human interaction will occur, consent form is not required from participants. Participants who qualify for the qualitative portion of the study will be provided with a detailed explanation of the study protocol. They will be informed that their participation is voluntary and that they may withdraw from the study at any time. Those who match the inclusion criteria and confirm their explicit verbal consent to participate in this study will be scheduled for an interview at a convenient time. Once the participant consents to participate in this study, they will be assigned a code indicating the chronological number of enrolments for disseminating the results while preserving anonymity. Their legal name and contact information will not be attached to the collected data. The study's quantitative and qualitative data will be kept in a secure location at the principal investigator's (first author) locked office at Lakehead University in a locked filing

**Table 3. Strategies to address trustworthiness of the study.**

| Criteria | Activity |
|---|---|
| Worthy topic | Given the burden of diabetes in Ontario and its life-threatening complications, this study is timely and relevant. |
| Rich Rigor | Sufficient data will be collected. In social constructivist studies, data saturation is often reached with 15–30 interviews. There will be synergy between methodology (social constructivism) and method (semi-structured interviews), and in-depth and systematic data analysis will be used. |
| Sincerity | We will engage in self-reflection practices through routine journaling to better understand our social position and the role of power and privilege in relation to participants and the context of the study. |
| Credibility | We will provide in-depth descriptions of participants' stories. We will engage in analyst triangulation (reflective commentary by researchers, independent analysis by more than one researcher), theory triangulation (using theoretical perspectives for interpretation of data), and member checking (participants will receive their verbatim interview to check for accuracy). |
| Resonance | We will present stories in an evocative manner. We will describe the setting, participants' characteristics, and methods of collection and analysis in detail to ensure other researchers can use our research questions and methods to investigate diabetes in different contexts. |
| Significant contribution | The results of this study will shed light on strategies needed to improve diabetes and diabetes-related outcomes, enhance self-management skills, wellness, health accessibility and equitable care among people living with diabetes in NWO |
| Ethical | Procedural ethics will be followed (ethics approval obtained, consent from participants), and protection of data will be through the university's secure service and accessible only to the research team. |
| Meaningful coherence | There is coherence between the research question/objectives, design, plan for data collection, data analysis and findings. |

cabinet. Only the research team gathering information will have access to this storage system. The authorized research assistants will enter study data into a secure electronic database stored at Lakehead University in the first author's office in Thunder Bay, ON, on a separate password-protected computer to prevent any person from accessing the data unless designated for that purpose.

## Research significance, impact, and outcomes

This study will contribute to: 1) understanding diabetes complications and social determinants of health that impact diabetes outcomes for individuals living with diabetes in a Northwestern community; 2) knowing the needs, concerns, and challenges that individuals with diabetes confront in monitoring and regulating their disease; 3) developing strategies needed to address the social determinants of health influencing outcomes of diabetes for the Northwestern Ontario population; and 4) planning for future research, policy, and health organizations' actions needed to improve diabetes-related outcomes in NWO communities, namely Thunder Bay.

In the short term, we expect to develop and strengthen partnerships with a diabetes care organization (i.e., CCDC) in Thunder Bay to develop current and future research to identify and address the impact of socio-economic determinants of heath on diabetes outcomes. In the long term, we aim to design and implement a future research intervention project that aims at developing, strengthening, and evaluating a diabetes self-care program to prevent diabetes-related consequences among Northwestern populations. A practical approach to chronic conditions requires a better understanding of the specificities of a multifaceted disease and the impact of its management on the health outcomes of the targeted population.

## Dissemination of research results

A knowledge translation plan informed by Farkas et al. [29] and Lavis et al. [30] has supported us to develop multiple strategies to disseminate research findings to a diversity of stakeholders include knowledge users. For instance, we have been engaging the healthcare team (i.e., knowledge users) of the research setting from the very beginning. Firstly, we met with the manager and a nurse practitioner of CCDC to present our initial research idea and allowed them to provide feedback. Secondly, we organized a few meetings over zoom with more members of the team to disseminate our first draft and gather their feedback on how we could support them with data to improve the diabetes program (i.e., CCDC). Finally, once we had the research proposal designed with all their input, we scheduled a time in their monthly meeting (October, 2022) to disseminate the main phases of the proposal in a Power Point presentation. The engagement of the CCDC team has been essential to improve each step of the project.

Once this study is completed, we plan to disseminate the research findings to the CCDC healthcare professionals (i.e., knowledge users) through a formal meeting. We also plan to publish the findings in peer-reviewed and open-access journals because they have the potential to reach a broader audience (e.g., epidemiologists, policy makers, researchers, and healthcare providers), and therefore, increase the likelihood of research uptake by the academic community, knowledge users, and the general public. Additionally, we plan to disseminate the research findings at the National and International Diabetes Conference to reach out to a variety of stakeholders and by using a non-academic mode of communication such as face-to-face meetings, workshop settings, and social media and webinars. Study's participants will be informed that the research results will be available upon request.

## Acknowledgments

The authors wish to thank the nursing staff of the Centre for Complex Diabetes Care (CCDC) in Thunder Bay, Ontario, for agreeing to assist with the recruitment of patients with diabetes for the study.

## Author Contributions

**Conceptualization:** Idevania G. Costa, Anna Koné.

**Data curation:** Idevania G. Costa.

**Formal analysis:** Idevania G. Costa, Pilar Camargo-Plazas, Anna Koné.

**Funding acquisition:** Idevania G. Costa.

**Investigation:** Idevania G. Costa, Kristen McConell, Kaitlin Adduono.

**Methodology:** Idevania G. Costa, Kaitlin Adduono, Pilar Camargo-Plazas, Anna Koné.

**Project administration:** Idevania G. Costa, Kristen McConell, Kaitlin Adduono.

**Resources:** Idevania G. Costa, Kaitlin Adduono.

**Supervision:** Idevania G. Costa.

**Validation:** Idevania G. Costa, Pilar Camargo-Plazas, Anna Koné.

**Writing – original draft:** Idevania G. Costa.

**Writing – review & editing:** Idevania G. Costa, Kristen McConell, Kaitlin Adduono, Pilar Camargo-Plazas, Anna Koné.

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
