## [Decision Letter · Decision Letter 0]

8 Feb 2023

PONE-D-22-20511Exploring diabetes status and social determinants of health influencing diabetes-related complications in a Northwestern Ontario - A mixed method study protocolPLOS ONE

Dear Dr. Costa,

Thank you for submitting your manuscript to PLOS ONE. After careful consideration, we feel that it has merit but does not fully meet PLOS ONE’s publication criteria as it currently stands. Therefore, we invite you to submit a revised version of the manuscript that addresses the points raised during the review process. Authors are suggested to strengthen their literature review of the topic, add strong rationale to the study, and justify the need to do mixed methods and the choice of parallel mixed methods design compared to others. Specific reviewers' comments are given below.

A rebuttal letter that responds to each point raised by the academic editor and reviewer(s). You should upload this letter as a separate file labeled 'Response to Reviewers'.A marked-up copy of your manuscript that highlights changes made to the original version. You should upload this as a separate file labeled 'Revised Manuscript with Track Changes'.An unmarked version of your revised paper without tracked changes. You should upload this as a separate file labeled 'Manuscript'

We look forward to receiving your revised manuscript.

Kind regards,

Sana Sadiq Sheikh

Academic Editor

PLOS ONE

Journal Requirements:

2. Please ensure that you include a title page within your main document. You should list all authors and all affiliations as per our author instructions and clearly indicate the corresponding author.

3**.** Your ethics statement should only appear in the Methods section of your manuscript. If your ethics statement is written in any section besides the Methods, please move it to the Methods section and delete it from any other section. Please ensure that your ethics statement is included in your manuscript, as the ethics statement entered into the online submission form will not be published alongside your manuscript.

Additional Editor Comments (if provided):

Please find the reviewers' comments below.

Reviewers' comments:

Reviewer's Responses to Questions

**Comments to the Author**

1. Does the manuscript provide a valid rationale for the proposed study, with clearly identified and justified research questions?

Reviewer #1: Yes

Reviewer #2: Yes

2. Is the protocol technically sound and planned in a manner that will lead to a meaningful outcome and allow testing the stated hypotheses?

Reviewer #1: Yes

Reviewer #2: Partly

3. Is the methodology feasible and described in sufficient detail to allow the work to be replicable?

Reviewer #1: Yes

Reviewer #2: Yes

4. Have the authors described where all data underlying the findings will be made available when the study is complete?

Reviewer #1: Yes

Reviewer #2: No

5. Is the manuscript presented in an intelligible fashion and written in standard English?

Reviewer #1: Yes

Reviewer #2: Yes

6. Review Comments to the Author

You may also provide optional suggestions and comments to authors that they might find helpful in planning their study.

Reviewer #1: 1. Please use same unique referencing format in whole the paper. For example, in the Introduction, line 5 you are suing two different formats. Same in the second page of Introduction, second paragraph line 7.

2. Please justify why you are doing both quantitative and qualitative studies parallel?

3. For the quantitative study, please explain what are the exclusion criteria?

4. For the qualitative study, what are the exclusion criteria?

5. For the qualitative study, please explain what you will do to increase trustworthiness such as credibility, transferability,…(study rigor).

6. Who will conduct the interviews?

Reviewer #2: It was a pleasure to review and learn about your work which I believe is very relevant to the research community as well as population health. While I found this a fair read, I have the following comments and considerations to work on and address.

General comments

Pay attention to claims without appropriate interpretation of reference

Provide reference citations for assertions that are not yours

A few grammatical and syntactical errors are replete in your piece. Address them

Ensure references are in alignment with PONE author guidelines (including dois where necessary, date of access for URLs, etc)

Specific comments

More context on the burden of DM on the local community in NWO compared to neighbouring areas would be necessary to understand and justify the research inquiry

“While, the focus of most studies has been on pharmacological and technical approaches to manage diabetes,” cite the studies that focus on pharmacological techniques. I beg to disagree because a cursory online scan of studies on DM in NWO shows a few studies focusing on DM epidemiology and outcomes.

“To date, no published study has explored the impact of gender, race/ethnicity and social determinants of health on diabetes-related complications for individuals living in Northwestern Ontario”.

As commented above, I disagree. Consider rephrasing as it is not totally true. There are a few studies albeit, localized or published over a decade. Often lazy scholars use the idea of ‘no evidence’ to rationalize study objectives when they failed to do a proper review of literature

“Northwestern Ontario (NOW)” do you mean NWO here.

“However, there is evidence that Canadian population with diabetes who are living in remote and rural areas of Northwestern Ontario are facing lots of challenges in addressing social and economic determinants of health, particularly food insecurity and access to services and resources to support their diabetes management” Provide the evidence by citing the source!

“Population groups that have more difficulty to access, services, resources, and fresh and healthy food due to financial constraints and low-income households, include those with lower levels of education, Indigenous peoples, and those living in remote regions [14]. Therefore, nonpharmacological interventions that addresses the social determinants of health and enhance selfcare behaviours may lead to prevention of diabetes-related complications improve quality of life and decrease the economic burden of disease [15], especially for people with Type 2 diabetes in rural and remote areas of Northwestern Ontario.” This is a valid point but the delivery could be re-structured to show the linkages between socio-personal determinants of health and food insecurity which I think is the point you trying to drive home here.

“However, the relationships between social economic factor and individuals’ attributes are not totally clear” what does literature say about this that is not totally clear? Include it, analysing for the gaps in literature and how your proposal will help bridge that gap. Don’t just provide vague and generic statements.

“Thus, in response to this gap, this research study will firstly use a cross-sectional design to identify the prevalence of diabetes complications, the internal and external factors affecting diabetes’ outcomes in a Northwestern Community. Secondly, it will employ a critical pedagogy approach to explore the needs, motivations and struggles individuals with diabetes face to selfmanage the disease on a daily basis.” This should be for the methods section. Move it.

Methods

“parallel mixed-method approach” it will be a good place to explain parallel mixed-method approach and why it best fits to answer your research questions.

The region of NOW has a significant amount of Indigenous communities of whom have higher inequalities socially, and clinically. I didn’t see how your proposal is designed to address Indigenous health from a community-led/engaged perspective. Indigenous peoples have higher rates of DM than non-Indigenous peoples. What engagement strategies, protocols and pursuits are in place to address inequalities in for this group of people?

I need more information on how the findings from this study would benefit patients with DM with regard to self-care, self-reliance, health promotion, and rehospitalization. Your focus is more on policy and population health but less on the individual scale.

Have included how your study will not add to increase health inequalities in other areas of chronic disease care and will address equity, diversity and inclusion of participants.

Comment on knowledge translation considerations as well. How will data findings be made available to relevant stakeholders

7. PLOS authors have the option to publish the peer review history of their article (what does this mean?). If published, this will include your full peer review and any attached files.

Reviewer #1: **Yes: **Masoud Mohammadnezhad

Reviewer #2: **Yes: **Udoka Okpalauwaekwe

---

## [Author Response · Author response to Decision Letter 0]

31 Mar 2023

March 21, 2023

To: Sana Sadiq Sheikh

Academic Editor 

Plos One Journal 

Re: PONE-D-22-20511 “Exploring diabetes status and social determinants of health influencing diabetes-related complications in a Northwestern Ontario - A mixed method study protocol”

Dear Dr. Sheikh,

We would like to thank you for allowing us to resubmit a revised copy of our manuscript and the reviewers for providing valuable comments and suggestions to improve it. We agree with most of the comments/suggestions made by the reviewers and have addressed them as outlined in the attached table. 

Thank you for the opportunity to review our manuscript; we believe that the reviewers’ thorough review and thoughtful comments have vastly contributed to improve its quality.

Sincerely,

The Authors

---

## [Decision Letter · Decision Letter 1]

28 Apr 2023

PONE-D-22-20511R1Exploring diabetes status and social determinants of health influencing diabetes-related complications in a Northwestern Community, Ontario, Canada: A mixed method study protocolPLOS ONE

Dear Dr. Costa,

Thank you for submitting your manuscript to PLOS ONE. After careful consideration, we feel that it has merit but does not fully meet PLOS ONE’s publication criteria as it currently stands. Therefore, we invite you to submit a revised version of the manuscript that addresses the points raised during the review process.

Please see the comments from the reviewers. Make necessary edits and send us back the revised manuscripts with changes. 

We look forward to receiving your revised manuscript.

Kind regards,

AKM Alamgir, PhD

Academic Editor

PLOS ONE

Journal Requirements:

Reviewers' comments:

Reviewer's Responses to Questions

**Comments to the Author**

1. Does the manuscript provide a valid rationale for the proposed study, with clearly identified and justified research questions?

Reviewer #1: Yes

Reviewer #2: Yes

2. Is the protocol technically sound and planned in a manner that will lead to a meaningful outcome and allow testing the stated hypotheses?

Reviewer #1: Yes

Reviewer #2: Yes

3. Is the methodology feasible and described in sufficient detail to allow the work to be replicable?

Reviewer #1: Yes

Reviewer #2: Yes

4. Have the authors described where all data underlying the findings will be made available when the study is complete?

Reviewer #1: Yes

Reviewer #2: Yes

5. Is the manuscript presented in an intelligible fashion and written in standard English?

Reviewer #1: Yes

Reviewer #2: Yes

6. Review Comments to the Author

You may also provide optional suggestions and comments to authors that they might find helpful in planning their study.

Reviewer #1: Thank you for considering all my comments. All the changes are made and the quality has promoted well.

Reviewer #2: Thank you for the opportunity to review your work again. While I commend your efforts to improve on the quality per reviewer comments, I have a few comments for considerations.

Confirm the following in your methods

Will data for the quantitative arm be collected `prospectively or retrospectively?

Would patient participants provide consent for their data to be collected via their charts?

Will participants be recruited first before data collection. Your order of proceedings makes it seem like they wouldn’t.

For the qualitative part

Who developed the interview guide? Were patients involved in the co-development (if that was done). Any field test of the guide for language, comprehension, etc?

Your KT planes seem rather vague here “We also plan to disseminate the research findings to participants, healthcare professionals, and knowledge users beyond the research community by using a non-academic mode of communication such as face-to-face meetings, workshop settings, and social media such as blogs, podcasting, and webinars. In this strategy our goal is to adapt the language of the publication to the target audiences by using plain language to help them understand the research findings at their level.”

Who has been engaged so far with this work you are doing? How would you engage them to use the findings. The aim of KT is beyond publication, but engagement to knowledge use and implementation. I would be interested to see what has been done with this piece so far and not what you plan to do in the future.

7. PLOS authors have the option to publish the peer review history of their article (what does this mean?). If published, this will include your full peer review and any attached files.

Reviewer #1: **Yes: **Masoud Mohammadnezhad

Reviewer #2: **Yes: **Udoka Okpalauwaekwe

---

## [Author Response · Author response to Decision Letter 1]

11 Jul 2023

We have attached a response to the reviewers with dated of July 6th, which address each comment.

---

## [Decision Letter · Decision Letter 2]

12 Sep 2023

Exploring diabetes status and social determinants of health influencing diabetes-related complications in a Northwestern Ontario - A mixed method study protocol

PONE-D-22-20511R2

Dear Dr. Idevania Costa,

We’re pleased to inform you that your manuscript has been judged scientifically suitable for publication and will be formally accepted for publication.

Kind regards,

AKM Alamgir, PhD

Academic Editor

PLOS ONE

Additional Editor Comments (optional):

Reviewers' comments:

Reviewer's Responses to Questions

**Comments to the Author**

1. Does the manuscript provide a valid rationale for the proposed study, with clearly identified and justified research questions?

Reviewer #2: Yes

Reviewer #3: Yes

Reviewer #4: Yes

2. Is the protocol technically sound and planned in a manner that will lead to a meaningful outcome and allow testing the stated hypotheses?

Reviewer #2: Yes

Reviewer #3: Yes

Reviewer #4: Yes

3. Is the methodology feasible and described in sufficient detail to allow the work to be replicable?

Reviewer #2: Yes

Reviewer #3: Yes

Reviewer #4: Yes

4. Have the authors described where all data underlying the findings will be made available when the study is complete?

Reviewer #2: Yes

Reviewer #3: Yes

Reviewer #4: Yes

5. Is the manuscript presented in an intelligible fashion and written in standard English?

Reviewer #2: Yes

Reviewer #3: Yes

Reviewer #4: Yes

6. Review Comments to the Author

You may also provide optional suggestions and comments to authors that they might find helpful in planning their study.

Reviewer #2: It was a pleasure reviewing your work again. Thanks for paying attention to the comments preferred. Wish you the best.

Sincerely

Reviewer #3: When we are considering the local population of northwestern Canada, why was an interpreter not made available? As I understand The native language is not English, to understand the practices for managing diabetes by the local population study should have included the non-English local population with the availability of an interpreter. This would help us to get more diabetics patients for our study.

Reviewer #4: Thank you very much for the opportunity to review your manuscript. The proposal appears sound, and thorough.

1. For demographic data collection - it might be beneficial to also collect data on racial/ ethnic background - specifically native/ immigrant backgrounds

It would be helpful to have a little bit more detail about social constructivist approach. Why is it better than other approaches. I understand that it may allow us to better explore the social lived experience of having diabetes, and how that impacts on their health, but more detail might be beneficial.

7. PLOS authors have the option to publish the peer review history of their article (what does this mean?). If published, this will include your full peer review and any attached files.

Reviewer #2: **Yes: **Udoka Okpalauwaekwe

Reviewer #3: **Yes: **Nishat Mehdi

Reviewer #4: No

---

## [Editor Report · Acceptance letter]

19 Sep 2023

PONE-D-22-20511R2 

Exploring diabetes status and social determinants of health influencing diabetes-related complications in a Northwestern Community, Ontario, Canada: A mixed method study protocol 

Dear Dr. Costa:

I'm pleased to inform you that your manuscript has been deemed suitable for publication in PLOS ONE. Congratulations! Your manuscript is now with our production department. 

Kind regards, 

on behalf of

Dr AKM Alamgir 

Academic Editor

PLOS ONE